# Assessing Natural Weaning in Suckler Beef Cattle: A Single-Farm Retrospective Data Analysis of Calf-Raising Success and Colostrum Antibody Uptake in the Absence or Presence of a Yearling Calf

**DOI:** 10.3390/ani16010034

**Published:** 2025-12-23

**Authors:** Dorit Albertsen, Peter Plate, Suzanne D. E. Held

**Affiliations:** 1Bristol Veterinary School, University of Bristol, Langford BS40 5DU, UK; 2Royal Veterinary College, Stinsford Business Centre, Kingston Maurward College, Dorchester DT2 8PY, UK

**Keywords:** natural weaning, suckler beef calves, animal welfare, sibling rivalry, raising success, colostrum antibody uptake

## Abstract

Commercial suckler beef calves are commonly physically separated from their dams at four to ten months old, often causing stress and health issues in cows and calves. This retrospective study looked at a ‘natural weaning‘ approach where cows kept their calves and therefore had a yearling still with them when the next calf was born. It investigated for the first time whether the presence of a yearling calf would be negatively linked to the raising success of the dam’s next calf. To that end, it compared the survival rates of a cow’s new calf when her yearling calf was still present with when the yearling was absent because it had been separated from her at 8–9 months. The comparison was thus between ‘natural weaning’ and conventional separation. This study was conducted on a single large, extensive, organic suckler beef farm on chalk downland in the south of the UK as it gradually changed its management to natural weaning. In a complementary study on the same farm, blood samples of newborn calves with the yearling sibling present or absent were tested for total protein levels to check for colostrum antibody uptake. Main findings are that leaving calves with their dams, and not separating them at 8 months old, is not associated with reduced raising success or colostrum antibody uptake for the subsequent calf. Natural weaning thus has potential as a management strategy for extensively reared suckler beef calves on comparable grasslands.

## 1. Introduction

Suckler beef calves are typically removed from their dams at four to ten months old [1,2,3,4,5]. This is earlier than the cow would naturally wean her calf off milk if left unmanaged [6] and is mostly carried out by physically separating calf and dam [2,7,8]. In calves, such abrupt separation before the age of natural weaning is known to trigger behavioural stress responses. These include increased calling, restless pacing interspaced with apathy-like states, reduced feed intake and consequently an increase in disease incidence due to stress-induced immune suppression, which cease when a reunion is possible [3,9,10]. Common consequences are outbreaks of “shipping fever”, affecting the calves’ respiratory tract [11], including subclinical forms that result in impaired weight gain [8,12,13,14,15]. Abrupt separation also increases physiological stress indicators such as heart rate and blood cortisol levels in the dams, and, again, these are reversed at reunion [16,17]. To address the negative consequences of early and abrupt separation, various alternative methods have been introduced. “Fenceline weaning”, for example, allows the cow and her calf physical contact across a barrier without being able to suckle for a period of time before full separation [18]. “Two-step weaning” fits calves with a nose flap that prevents them from suckling but allows full physical contact with their dam while adjusting to a diet without milk [19]. Both methods reduce stress behaviours such as vocalisation and pacing, increase the time spent eating and lying and improve calf health and performance indicators [3,8,18,19]. Our paper for the first time presents calf survival data for a further method towards the same aim of finding viable alternatives to early, abrupt cow–calf separation in beef suckler herds. The method introduced here is natural weaning. ‘Natural weaning’ means cows were left to wean their calves by themselves. Consequently, yearling calves stayed with their dam beyond the subsequent calving season and were still with her when her next calf was born. Retrospective survival data were collected on the new calves born in that subsequent calving season. The data analysis was carried out on a single large extensive suckler beef farm in England where conventional cow–calf separation was gradually phased out over a period of seven years, to be fully replaced by natural weaning. The survival of the newborn calf and successful raising to weaning depend on the dam’s ability to provide immediate immune protection via colostrum and then raise it, that is, nurse and care for it. The cow and her new calf must establish and maintain a bond that motivates the cow to accept her calf and feed and protect it [20,21,22]. A yearling sibling, present at and beyond the new calf’s birth, might negatively affect the dam’s success in raising the new calf in a number of ways [23,24,25]: by disturbing the bonding process, which could cause the cow to not prioritise care for her newborn or even abandon it [9,26,27]; by “stealing” colostrum produced for the new calf before and after its birth [28,29,30]; and by competing with the new calf for milk, which could leave it malnourished [24,31,32]. Cows have a cotyledonary synepitheliochorial placenta that does not allow for passive transfer of immunoglobulins from dam to calf, and consequently calves are born immunoincompetent. Therefore, adequate intake of colostrum after birth is vital for newborn calves and other mammals with a similar type of placenta [33]. It is essential that the cow nurses no other calf than her newborn before and after its birth and that her newborn is undisturbed by others while learning to suckle: the ingestion of adequate amounts of colostrum within the first 12–24 h of life is vital for absorbing protein macromolecules, including immunoglobulins, as the neonatal small intestine will only absorb these in the first hours postnatum to gain passive immunity against infection [34,35,36]. In addition to providing this passive immunity, the colostrum contains hormones and growth factors and promotes the calf’s metabolism [34,35]. Serum immunoglobulin G (IgG) levels, as originate from colostrum, predict the risk of dying or requiring treatment in the newborn suckler calf. For every 5 g/L increase in serum IgG, the odds ratio of dying and/or requiring treatment decreases by 0.86 [30]. Beyond colostrum, the amount of milk available to the new calf furthermore strongly influences its growth rate and survival in the first year of life [37,38,39]. As one performance variable, we therefore analysed farm data on serum total protein (sTP) levels in the first days of the new calf’s life to assess the potential effect of a yearling’s presence on its colostrum antibody intake. Serum total protein, measured with a refractometer, has been established as a reliable indirect measure for the concentration of IgG in neonatal calves up to 10 days of age [40]. The second performance variable assessed the potential effect of a yearling calf on the dam’s ability to continue to nurse and raise a newborn calf successfully to 12 months old (‘raising success’).

This study presents for the first time an analysis of calf-raising success in an extensive suckler beef system during its transition from conventional cow–calf separation to natural weaning. The focus is on the cow’s ability to raise her calf in the presence of her older offspring, as a yearling may still be suckling when the dam is getting close to giving birth to her next calf. This raises the question of whether and how the continued presence of her previous, now ’yearling’, calf affects a dam’s success in raising her subsequent calf. Raising success rates are compared between cows with their yearling calf either present or absent beyond the subsequent parturition. Raising here is considered successful when the calf reaches nutritional independence. Independence requires a fully functional rumen and forage of a quality that allows the calf to grow. In a spring calving herd at pasture, both requirements are met at one year of age, which was therefore used as the definition of raising success. The colostrum antibody uptake of calves born in the presence of their yearling sibling is also investigated and, again, compared to calves born in the absence of an older sibling.

## 2. Materials and Methods

### 2.1. Study Animals, Environmental Conditions and Animal Management

Retrospective data were collected on an extensive commercial beef suckler farm in southern England. The farm rents 3300 hectares of chalk downland in southern England. Altitude ranges between 120 and 200 m above sea level with latitude at 51° N and longitude at 2° W. The climate is mild with an average temperature of 9.4 °C and an annual rainfall average of 755 mm. The land is designated as a Site of Special Scientific Interest (SSSI), with a main focus on protecting ground-nesting birds and insects and conserving species-rich chalk grassland. No reseeding or fertiliser application takes place. The farm’s cattle graze this native pasture throughout the year with additional water and a mineral lick provided ad libitum. The farm is certified as organic and sells grass-finished beef.

The study sample consists of 1822 new calves born to 663 cows, all as part of the farm’s commercial stock. The stock typically comprises between 1400 and 1800 beef cattle in total, depending on the season: numbers increase during the calving period in April, May and June, then decline again as fat stock and store cattle are sold, mostly in late summer and autumn. The 663 study cows were predominantly Aberdeen Angus and included 50 White Park cows kept for nature conservation purposes. They lived in nine separate breeding herds of between 30 and 200 animals that were established and managed by retaining female offspring as replacement cows. Each herd was assigned a home range and kept in it by electric fencing, which was moved twice a month within the home range. Cow ages ranged from two to 15 years. Young stock that had not been separated in the conventional way at eight months old remained in the herd for 18 months if male. If female, they remained in the herd for 15 months or for their entire reproductive life. Bull calves were castrated in their first week of life. Thirty breeding bulls, aged one to nine years, joined the herds in late June for four months in groups of up to four, depending on herd size. The rest of the year the bulls lived in their own pastures in groups of two to twelve. One herd retained their breeding bulls permanently. The age at first calving was most commonly two years.

### 2.2. Raising Success Data Collection and Analyses

Retrospective ‘raising success’ data were collected for 1822 liveborn calves between 2006 and 2012 born to 663 cows while the farm’s breeding herds were one by one converted to natural weaning. These were all the cows who gave birth to a live-born calf in at least two consecutive years. Data were analysed from farm records. Each of the 1822 calvings was categorised as either with the cow’s yearling calf still present because it was left for natural weaning (YP), or absent (YA) because it was conventionally removed at about 8 months old. ‘Raised’ (R) was defined as the new calf being alive and present at its first birthday; ‘not raised’ (NR) was that it was not (it had died or was taken away because it was at risk of dying without management intervention). Only the raising of a live-born calf was considered, excluding calvings of stillborn calves.

Each of the 1822 newborn calves in the data analysis thus belonged to one of four categories: raised successfully to one year old in the presence of a yearling sibling (YPR); raised successfully in the absence of a yearling sibling (YAR); not raised successfully in the presence of a yearling sibling (YPNR); not raised successfully in the absence of a yearling sibling (YANR). Data were analysed using 2 × 2 Chi-squared tests of independence to check associations between yearling presence and subsequent calf-raising success. Data are first presented for all live calvings across years (i) and then statistically analysed by calendar year (ii), cow age (iii) and for each cow’s penultimate calf (iv), with (ii), (iii) and (iv) avoiding repeated calvings of the same cow in the analyses.

### 2.3. Serum Total Protein Collection and Analyses

In a separate study, existing on-farm data on serum total protein levels of 93 newborn calves from two of the farm’s herds during one calving season were analysed. At the time of blood sampling, calves were between one and ten days old, born to 93 different cows. A total of 81 of the cows still had their yearling present (YP), and 12 did not (YA), either because this was their first calf (N = 2) or they had not calved the previous year (N = 10). Blood samples were taken at the time of ear tagging as part of routine veterinary herd surveillance once per calf over a 6-week period from early April to mid-May 2024. Serum total protein was used as an indirect measure of IgG as previously validated in calves up to 10 days of age. Total protein levels in serum were measured via standard optical refractometry [41]. To establish whether the presence of the yearling affected the current calf’s colostrum antibody uptake, we first compared total protein levels against categories established for dairy calves [42] and tested for associations between category level and yearling presence using 2 × 2 Chi-squared tests; we also compared total protein between YP and YA calves using an Independent-Samples Mann–Whitney U test after checking for normality (Shapiro–Wilk test).

We then established the 12-month survival of this subset of 93 calves and used a Chi-squared test to investigate a potential association with the presence of the yearling and the colostrum uptake category.

All statistical tests were carried out using IBM SPSS Statistics, version 30.0.0.0. 

## 3. Results

### 3.1. Raising Success

#### 3.1.1. Total Number of Live-Born Calves Across All Years

Raising success totalled over the six years was 95.9% for calves born with the yearling calf present and 94.7% for calves with the yearling calf absent, reaching one year of age (Figure 1). Each cow contributed at least two live-born calvings to the total number. A total of 1005 of these calves were born in the presence of a yearling (YP); of these, 964 were successfully raised by their mothers to one year old. A total of 817 calves were born without a yearling present (YA); of these, 774 were successfully raised.

#### 3.1.2. By Calendar Year

The figures per calendar year are summarised in Table 1, showing no major numerical variations in calf survival between calendar years.

#### 3.1.3. By Age of the Dam

Raising success rates by cow age is illustrated in Figure 2. Total numbers of live-born calves for whom the age of the dam was available were N = 549 for 3-year-old cows, N = 403 for 4-year-olds, N = 282 for 5-year-olds, N = 233 for 6-year-olds, N = 146 for 7-year-olds, N = 93 for 8-year-olds, N = 57 for 9-year-olds, N = 31 for 10-year-olds and N = 23 for 11-year-olds.

Too few data were available for Chi-square testing when cows were between 6 and 10 years old because cases were fewer than five per year where calves were not successfully raised with either the yearling present or absent. At the ages of 3–5, tests revealed no significant association between raising success and yearling presence. Figure 2 also shows a numerical rise for calf-raising success rates to increase with age up to 10 years old, whether with or without a yearling present. Changes with age were not statistically analysed because of the amount of missing data from individuals.

#### 3.1.4. The Cows’ Penultimate Calf

When only one live-born calving per cow was considered (n = 663), with the penultimate calving chosen for standardisation, this split into 382 cows calving in the presence of their yearling (YP) and 281 cows calving with their yearling absent (YA). Of these 663 calves, 22 YA and 22 YP calves failed to survive to 12 months old. A total of 360 (94.27%) cows thus successfully raised their newborn calf in the presence of their yearling, and 259 (92.17%) cows successfully raised their calf in its absence. Chi-square test analysis found that raising success in this sample was not associated with yearling presence (Χ^2^ = 1.120, N = 663, df = 1, *p* = 0.29).

### 3.2. Serum Total Protein Levels

#### 3.2.1. Comparison Based on Agreed Categories for Dairy Calves (Lombard et al., 2020) [42]

Total serum protein levels for all 93 sampled calves are summarised in Table 2 and Figure 3. Eighty-one (87%) of them had ‘excellent’ protein levels. A Chi-squared test of independence found no association between yearling presence and the level of total serum protein in the new calves (categories amalgamated into excellent/good and fair/poor; Chi-square = 0.256; df = 1; *p* = 0.61).

#### 3.2.2. Comparison Based on Serum Total Protein as a Continuous Variable

Serum total protein levels across the 93 samples showed a trend towards higher serum total protein levels in calves with the yearling absent, just outside the level of significance (U = 318.0; *p*= 0.054). Figure 3 shows the distribution of values in YP and YA calves.

## 4. Discussion

This retrospective study investigated the impact of natural weaning on calf survival and colostrum antibody uptake. The growth and survival of a new calf might be compromised by competition for the dam’s milk and care if the yearling calf is still present. This would be the case in a natural weaning system where older calves are left with their mothers to be weaned naturally if the cow fails to wean. The comparison presented here is between beef calves raised in an extensive commercial suckler herd either in the presence or absence of their yearling sibling. It found no negative effect of sibling presence on calf survival to 12 months old, nor on colostrum antibody uptake.

### 4.1. Raising Success

Raising success rates in this sample of cows were 96% success overall in the presence of the yearling and 95% without a yearling present. They varied from 90% to 100% in cow age cohorts up to 10 years old. Analysis of only one—her penultimate—calf per cow similarly did not reveal a negative impact of yearling presence on calf survival to 12 months old.

Raising success might be expected to be lowered in the presence of a yearling for several reasons.

Firstly, the cow and her newborn must be undisturbed to form a bond. Secondly, the calf must learn to suckle and receive its colostrum while undisturbed. These situations could be negatively impacted by a yearling calf demanding the cow’s attention. Thirdly, the amount of milk available to the calf determines its growth rate in its first year [37,39]; malnourished young struggle to thrive and are susceptible to disease [43]. Having a potentially unweaned, older sibling present would introduce competition for the dam’s milk; that is, it would introduce sibling rivalry, which could leave the new calf malnourished. Such competition has long been recognised as likely to disadvantage younger offspring. Martínez-Gómez et al. [44], Trillmich and Wolf [32] and Hudson and Trillmich [31] described this as a general phenomenon with negative effects on the survival of neonate mammalian offspring. Against this prediction, Veissier et al. [5] reported for cattle that the yearling’s presence did not negatively influence the relationship between the cow and the newborn calf. In that study, the cows grazed in the yearling’s company while giving more direct care to the young calf, with four out of eleven yearling calves still suckling their dam after the birth of their younger sibling. Reinhardt [45] reported that 1 out of the 132 yearling calves investigated also did this. Note, however, that neither of these two studies followed newborns to 12 months old, and different findings were reported by Murphey et al. [46] for farmed water buffalo. In their study, frequent nursing of other calves depressed growth rates of the mother’s own calf. These calves were unrelated competitors stealing milk from mothers other than their own [46]. Even though the thieves were unrelated, the water buffalo study demonstrated the potential for a negative impact of nursing a second calf with the cow’s current one. We found no similar negative influence of a sibling on survival to 12 months of the cow’s current calf in the present study. Instead, our findings are in line with the previous reports above of cows coping well with an additional, older calf. They were able to bond with their newborns when older calves were present, and they supplied them with sufficient milk to survive to one year old. One possible explanation is that cows in the present study were able to compensate for the additional milk demand, if they had not weaned the yearling, simply by increasing their yield [47]. De Rose & Wilton [48], for example, reported suckler beef cows that were rearing twins increasing their milk yield by 20–40%. Another mechanism counterbalancing sibling competition could be the lower suckling frequency in older calves if not weaned. This would still provide the younger calf with a sufficient proportion of uncompromised sucking bouts [6,27,49,50,51,52,53,54,55,56]. Note that no conclusions can be drawn on actual milk intakes or growth rates from our data, as only survival rates were available, nor on whether the yearling calves were indeed still drinking milk or had been nutritionally weaned before the birth of the next calf but were still present. Furthermore, whether the presence of an older sibling specifically affects growth rates in a new calf in naturally weaning suckler beef herds also warrants further studies.

Another reason for poorer calf-raising success with yearling presence could be maternal inexperience or age. Three years is the youngest age at which a cow might simultaneously attend to and/or nurse a yearling (her first ever calf) and a newborn (her second). The data suggest a trend for mature cows to have higher raising success rates than younger cows, independent of yearling presence. Three-year-old cows showed a trend towards having a lower raising success rate in the presence of their yearling. All the three-year-old cows in this study were second calvers, meaning that they experienced the presence of more than one offspring as a novelty. The trend towards an age effect found here is in line with biological predictions and previous findings. The level of a dam’s experience is known to influence her maternal behaviour [57], as reported by Dwyer and Lawrence for sheep [58] and Edwards and Broom for cattle [20]. Edwards [59], for example, reported primiparous cows to occasionally fail to lick their calf, a lack-of-care phenomenon not observed in multiparous ones in the same study. Experience thus plays a part in bonding and calf rearing. This becomes especially important under natural weaning systems as studied here, when cows must cope with the presence of a yearling still demanding their attention at the time of their subsequent parturition. Murphey et al. [46] found that lack of maternal experience in young water buffalo cows made them targets for milk “theft” by hungry, non-filial calves. These calves knew which cows were worth approaching for allo-suckling and targeted the inexperienced, young ones during their own calves’ suckling bouts. In cattle, Edwards [59] similarly found second calvers to be the most subjected to allo-suckling, again suggesting that lifetime experience plays a role. Lack of experience might have made younger cows targets for milk “theft” by their own yearlings. Drewry et al. [60] found older beef cows to be heavier milk producers, with the maximum achieved by cows soon after maturity was reached. In the present study, raising success was high in the presence of a yearling calf, even in the younger cows. Relative inexperience in young cows in rearing a new calf with an additional yearling calf present, and a more limited capacity to compensate for increased milk demand with increased yields in the potential case of an unweaned yearling, are likely explanations for the insignificant trend towards higher raising success when study cows were older.

It has been suggested that having siblings stay with the dam beyond being fully dependent on her may carry benefits as well as costs. Matrilinear family groups underlie herd structures in many species and are considered influential for the formation of coalitions between individuals, as reported for cheetahs [61], baboons [62,63], lions [64], carnivores [65] and cattle [66]. Cattle preferentially associate with related conspecifics and invest more into social relationships with kin; when able to develop and maintain natural social bonds, domestic bovines develop natural herd structures of related females and their offspring [67,68,69,70]. Heck [71] reported that American bison calves stay with their dam for almost a year and are pushed out of her immediate proximity shortly before the birth of her next calf. However, Krasińska & Krasiński [72] found young European bison calves suckling in opposite parallel positions at the same time the previous year’s calves were suckling from behind. European bison stay with their dam through their first year and remain in the same group in their second year. Consequently, they are present at their dam’s subsequent parturition. Both American bison and Zebu cows maintain a close relationship after weaning with their calves [6,73]. Females of various ruminant species stay in their home herd (Carmargue cattle [26], red deer [74]). In the extensively managed suckler beef herd also investigated here, Johansen [67] had similarly reported social relationships among related cows to be stable throughout the start of the weaning period and that the presence of yearling calves did not influence successful establishment of the dams’ bond with their next calf; and we report here a small, insignificant trend for cows to have higher calf-raising success rates in the presence of their yearling for most cow ages (see Figure 2).

### 4.2. Colostrum Antibody Uptake

Colostrum antibody uptake as indirectly measured by serum total protein was used here as a second calf performance indicator. Serum total protein, measured with a refractometer, is a reliable indirect measure of serum IgG concentration in neonatal calves. The method was validated by Elsohaby et al. [41] and Wilm et al. [40]. Beef cows tend to have a lower volume of colostrum but a higher IgG concentration than dairy cows [75], as milk yield is negatively correlated to colostrum quality. Consequently, the required volume of intake is lower in beef cows. Other established factors affecting colostrum quality in dairy cows are age of the dam, dry period length and time from birth to first sucking/milk harvest [76], but research in beef cows is very limited [75].

Adequate colostrum intake has been associated with decreased preweaning morbidity and mortality [77], improved weight gain, reduced age at first calving and higher milk yields in the first and second lactation in dairy cows [78]. In beef cows, lower calf morbidity and mortality and higher growth rates were observed in calves with higher serum IgG concentrations [30].

In the present study, 87% of serum total protein levels across the 93 study calves were classified as ‘excellent’ (using the classification for dairy calves by Lombard et al. [42]), yielding better results than reported elsewhere for suckler calves. Bragg et al. [79] had found only 63% of 1131 calves from 84 farms in Great Britain (mean herd size 132 cows, range 20–1100) had serum IgG levels to be classified as excellent (over 24 g/L, equivalent to about 6.1 g/dL serum total protein in a study). The results of the present study demonstrate that effective colostrum management can be achieved despite the yearling calf being present in most of the cow–calf pairs. Note that total protein values were insignificantly higher in calves from cows without yearlings, but the health implications for the calf were negligible because the vast majority of sTP values in both groups were in the “excellent” category.

The classification for dairy calves was chosen, as no commonly agreed classification for beef calves exists.

### 4.3. Limitations of This Study

This study was retrospective, using entirely existing farm data collected over the years, and more information could be gathered in a prospective, randomised controlled trial. No data were available regarding disease incidence, weights and growth rates in the calves, which should be included in further studies of this type. The reliance on existing data also led to some small sample sizes, especially the YA group in the colostrum antibody uptake study. The effects on subsequent suckler cow fertility were also not part of this study and require further analysis.

## 5. Conclusions

The aim of this study was to assess the suitability of natural weaning for extensively managed suckler beef cattle based on two performance measures: raising success and colostrum antibody uptake of the next calf. Its main finding is that cows achieved high performance on both measures, independently of the presence or absence of a yearling calf. It is the first demonstration of natural weaning working well in a suckler beef management system such as the one studied here. Because this study was conducted under one set of environmental and husbandry factors only, its findings are proof-of-concept with limited generalisability at this point. That said, because of the large number of calvings available for analysis, the findings provide a solid basis from which to recommend further trials of natural weaning in beef cattle. While overall levels of raising success were high, the data nonetheless suggested a small age effect that would need to be confirmed with a view to mitigation if problematic. Not touched upon in this study were growth rates in the new calves nor reasons for non-success, that is, for calves not surviving to 12 months old; for example, disease, non-thriving, abandonment and accidents. Likewise, little is known about the behavioural weaning process as it naturally progresses between the cow and her yearling calf. A better understanding of underlying bio-behavioural factors such as these, together with agro-economic modelling, will support wider application of natural weaning in domestic cattle. Building on natural weaning, our findings point to the possibility of rearing and keeping commercial suckler beef cattle in herds of family groups as an alternative to management that relies on cow–calf separation. This should improve welfare by reducing separation stress and health problems in cows and weaned calves and by enabling young cattle to mature in natural, age-structured social groups.

Knowledge of unmanaged, ‘natural’ social dynamics and bonds beyond weaning in cattle is not solely important on grounds of cow and calf welfare. Domestic cattle are now also looked to as playing a key role in nature restoration, standing in for their missing ancestors, the aurochs and other megafauna. By enhancing biodiversity, they support habitat and soil regeneration [80]. In areas that employ the hands-off practice known as rewilding to achieve nature rehabilitation, these cattle must be capable of thriving with minimal human intervention, with their welfare and survival safeguarded. This, too, can be supported by a stable social structure and a herd composition where animals can build beneficial long-term relationships that protect the younger individuals and allow for the passing on of experience in the native environment [81,82,83,84]. Experience with leaving suckler beef cattle with their family in their herd of birth suggests that sibling rivalry does not pose a problem and that raising cattle in naturally grown herds is a viable solution that could enhance animal welfare while supporting young individuals in wild grazing and habitat regeneration projects.

## Figures and Tables

**Figure 1 animals-16-00034-f001:**
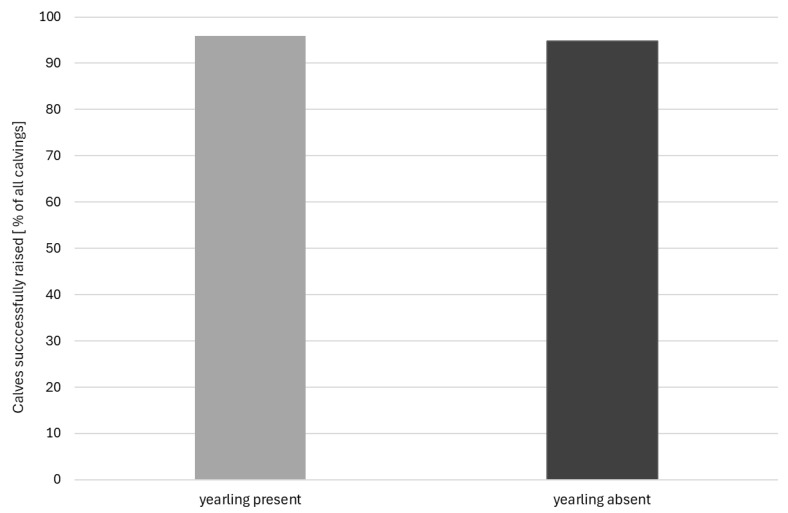
Raising success percentages for the total number of live-born calves over the six study years (N = 1822).

**Figure 2 animals-16-00034-f002:**
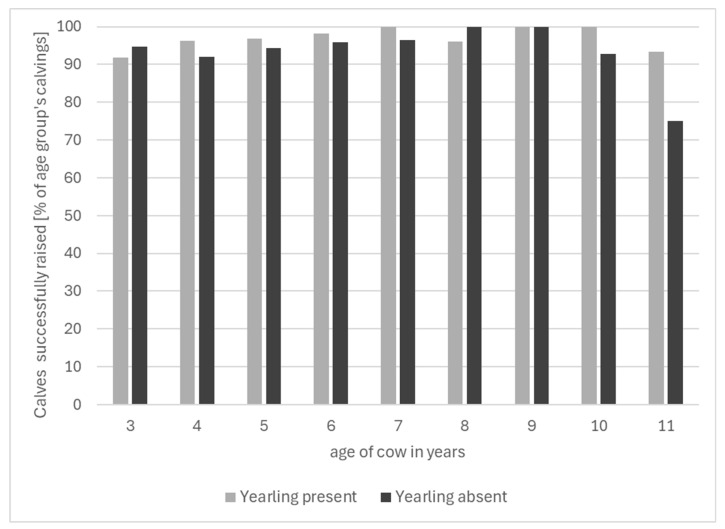
Raising success percentages of live-born calves by cow age over the whole of the observation period; total numbers of live-born calves per age are given in the text.

**Figure 3 animals-16-00034-f003:**
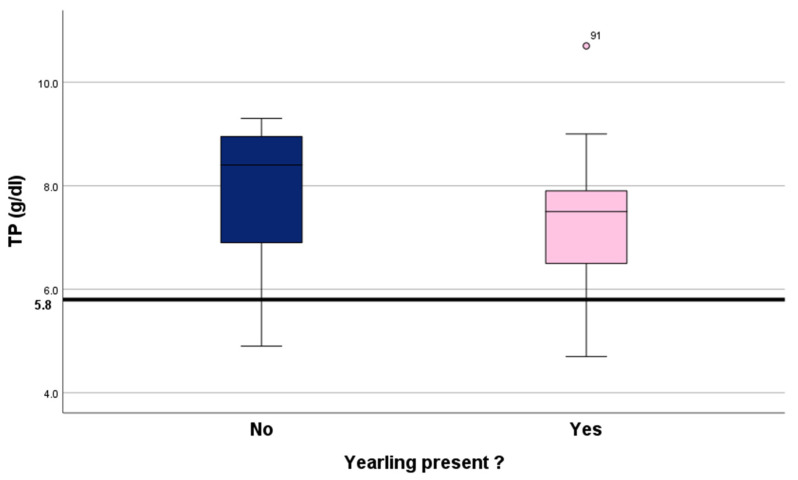
Distribution of serum total protein levels depending on the presence of the yearling. Yearling absent: n = 12, yearling present: n = 81. Given are medians and 1st and 3rd quartiles. The 5.8 g/dL line is the cut-off point for ‘good’ colostrum antibody uptake according to the classification for dairy calves by Lombard et al. [42].

**Table 1 animals-16-00034-t001:** Calf-raising success rate per year for cows with their yearling present or absent (n = 1822 total calves from 663 cows across the years).

Year	YPR Yearling Present, Raised	YAR Yearling Absent, Raised	YPNR Yearling Present, Not Raised	YANR Yearling Absent, Not Raised	% Raising Success with Yearling Present	% Raising Success with Yearling Absent
2006	10	98	0	1	100	99
2007	52	109	2	0	96.3	100
2008	99	141	6	10	93.4	93.4
2009	161	75	6	4	95.3	94.9
2010	185	73	13	2	93.4	97.3
2011	199	160	4	11	98	93.6
2012	258	118	10	15	96.3	88.7
TOTAL	964	774	41	43	95.9	94.7

**Table 2 animals-16-00034-t002:** Classification of serum total protein levels in young calves with a yearling present or absent (based on Lombard et al. [42]).

Total Protein	Category	Yearling Present (%)	Yearling Absent (%)

≥6.2 g/dL	Excellent	70 (86.4%)	11 (91.7%)
5.8–6.1 g/dL	Good	0 (0.0%)	0 (0.0%)
5.1–5.7 g/dL	Fair	10 (12.3%)	0 (0.0%)
<5.1 g/dL	Poor	1 (1.2%)	1 (8.3%)

## Data Availability

The original contributions presented in this study are included in the article. Further inquiries can be directed to the corresponding author.

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
