# Peer review of "Assessing Natural Weaning in Suckler Beef Cattle: A Single-Farm Retrospective Data Analysis of Calf-Raising Success and Colostrum Antibody Uptake in the Absence or Presence of a Yearling Calf"

_animals, 2025, doi:10.3390/ani16010034_

Round 1

Reviewer 1 Report

Comments and Suggestions for Authors

The manuscript presents a valuable and unique dataset from a 7-year field study on natural weaning in suckler beef cattle. The topic is highly relevant to animal welfare and sustainable production. However, major methodological and analytical issues must be addressed before publication.

Major Comments:

1. The sample size for the yearling-absent (YA) group (n=12) is disproportionately small compared to the yearling-present (YP) group (n=81), compromising statistical power and raising concerns about sampling bias.

2. Chi-square tests were not conducted for cows aged 6–10 years due to <5 unsuccessful raising cases per year, omitting critical insights into mature-to-elderly cow performance. Additionally, the trend of lower raising success in 3-year-old cows (second-calvers) lacks behavioral or physiological validation.

3. The Informed Consent Statement only mentions "permission from the farm owner" and omits approval from an institutional animal ethics committee—a requirement for animal research published in Animals.

4. Line 116-118: "Raising success" is defined solely as "survival to 1 year," with no inclusion of growth metrics (e.g., weight, daily gain)—key indicators of calf productivity.

5. Line 210-214: Chi-square analyses showed no association between yearling presence and raising success in 2008 (χ²=0.09, p=0.77) but a significant positive association in 2012 (χ²=8.67, p=0.003). The study fails to explain drivers of this discrepancy.

6. Table 2 shows no calves in the "Good" category (5.8–6.1 g/dl) for serum total protein, with no explanation for this absence—readers may question the validity of the classification scheme.

7. The study hypothesizes that "milk production compensation by cows" and "low suckling frequency of yearlings" explain the absence of negative effects, but no data on milk yield or suckling behavior are provided.

8. The conclusion claims natural weaning is "a management option for similar beef suckler herds," but the study is limited to a single farm (extensive organic management on chalk downland in southern England).

9. The study only mentions "unmeasured growth rates" and "unidentified calf mortality causes" as limitations, ignoring critical gaps such as "impact of natural weaning on yearling calf performance" and "long-term effects on cow calving intervals".

Minor comments:

1. Table 1 (Calf raising success rate per year) has incomplete column headers (e.g., "YPNR year-," "yearling pre-") and missing sample size labels, reducing readability.

2. The abstract incorrectly spells "cotyledonary synepitheliochorial placenta" as "cotolydonary synepitheliochorial placenta"—a critical anatomical term.

Author Response

Dear reviewer, thank you very much for your comments, please find below our responses and the re-written paper. 

Reviewer 2 Report

Comments and Suggestions for Authors

This manuscript addresses an underexplored management question and presents data from a unique long-term commercial setting with regard to "natural weaning." In that sense, it offers a useful glimpse into how natural weaning may function under extensive, low-input conditions. At the same time, several limitations temper the strength of the inferences that can be drawn (including the retrospective design, the lack of control for confounders such as parity and year effects, and the relatively small sample size for some measures). The observational nature of the work means that causal conclusions cannot be firmly established, which the authors acknowledge.

I was particularly disappointed too see that cow reproductive performance, longevity, BCS, and stayability were not robustly analyzed with respect to natural weaning. Nevertheless, the authors do acknowledge the limited inference space of their findings and do report these results as rather preliminary, case-study-esque observations that could inform designs of more robust experiments. This is appropropriate. Thus, despite some limitations, the study provides a somewhat useful case report that may help motivate more rigorous future research on alternative weaning strategies. I commend the authors for a well written and well-organized publication that cites appropriate relevant literature. 

Author Response

(The authors gave the same response as above.)

Round 2

Reviewer 1 Report

Comments and Suggestions for Authors

Thanks for the authors to address all comments in this revised version. Please revised the format of cited references in the manuscript.

Author Response

Thank you for your comments. We have changed the formatting of the references and corrected a couple of minor issues in the text. All changes from the previous version are marked. We believe all issues have now been addressed.